# In situ atomistic insight into the growth mechanisms of single layer 2D transition metal carbides

Xiahan Sang [1], Yu Xie [1], Dundar E. Yilmaz [2], Roghayyeh Lotfi[2], Mohamed Alhabeb [3], Alireza Ostadhossein [2], Babak Anasori[3], Weiwei Sun [1], Xufan Li[1], Kai Xiao[1], Paul R.C. Kent [1,4], Adri C.T. van Duin[2], Yury Gogotsi[3] & Raymond R. Unocic [1]

Developing strategies for atomic-scale controlled synthesis of new two-dimensional (2D) functional materials will directly impact their applications. Here, using in situ aberration-corrected scanning transmission electron microscopy, we obtain direct insight into the homoepitaxial Frank–van der Merwe atomic layer growth mechanism of TiC single adlayers synthesized on surfaces of $Ti_3C_2$ MXene substrates with the substrate being the source material. Activated by thermal exposure and electron-beam irradiation, hexagonal TiC single adlayers form on defunctionalized surfaces of $Ti_3C_2$ MXene at temperatures above 500 °C, generating new 2D materials $Ti_4C_3$ and $Ti_5C_4$. The growth mechanism for a single TiC adlayer and the energies that govern atom migration and diffusion are elucidated by comprehensive density functional theory and force-bias Monte Carlo/molecular dynamics simulations. This work could lead to the development of bottom-up synthesis methods using substrates terminated with similar hexagonal-metal surfaces, for controllable synthesis of larger-scale and higher quality single-layer transition metal carbides.

---

[1] Center for Nanophase Materials Sciences, Oak Ridge National Laboratory, Oak Ridge, TN 37831, USA. [2] Department of Mechanical and Nuclear Engineering, The Pennsylvania State University, University Park, PA 16802, USA. [3] Department of Materials Science and Engineering, and A.J. Drexel Nanomaterials Institute, Drexel University, Philadelphia, PA 19104, USA. [4] Computational Science and Engineering Division, Oak Ridge National Laboratory, Oak Ridge TN 37831, USA. These authors contributed equally: Xiahan Sang, Yu Xie. Correspondence and requests for materials should be addressed to X.S. (email: sangx@ornl.gov) or to R.R.U. (email: unocicrr@ornl.gov)

The last decade has witnessed rapid exploration of two-dimensional (2D) materials for a wide range of applications and across many fields due to their unique properties[1–3]. 2D transition metal carbides (TMCs) such as MXenes are promising candidates for various applications including electromagnetic interference shielding[4], energy storage[5–8], superconductors[9], catalysis[10], optoelectronics[11], sensors[12], medicine[13], and electrodes for 2D electronics[14]. MXenes are a large family of 2D materials[6,15,16], where 'M' is a designation for an early transition metal and X is carbon or nitrogen. Similar to graphene and transition metal dichalcogenides (TMDs), synthesis of TMC is accomplished using either bottom-up (e.g., chemical vapor deposition (CVD))[9] or top-down methods (e.g., chemical exfoliation)[17,18]. Compared to top-down methods, bottom-up synthesis enables growth of large-scale, high-quality 2D materials while also providing increased opportunity to tailor heterogeneity for electronic and optoelectronic applications[19,20]. However, until now, bottom-up synthesis of single layer TMC has not been reported experimentally, although theory has predicted such possibility[21]. CVD has been utilized to produce ultrathin $Mo_2C$ and other carbides, but their thicknesses were larger than 3 nm[9,22]. To develop large-scale synthesis methods for atomically thin TMC layers, exploring suitable substrates and understanding the interaction between substrate and TMC are of paramount importance.

In this work, the in situ homoepitaxial growth of hexagonal TiC ($h$-TiC) TMC single-layer flakes on monolayer $Ti_3C_2$ substrates, activated by combined thermal energy and electron-beam irradiation at 500 °C, and solely by thermal energy at 1000 °C, is investigated with atomic resolution using in situ scanning transmission electron microscopy (STEM), providing direct experimental evidence of the Frank–van der Merwe (FM) growth mode of single-layer TMC[23], analogous to molecular beam epitaxial (MBE) growth of 2D materials. The source atoms sustaining homoepitaxial growth are Ti and C adatoms that migrate from $Ti_3C_2$ flakes onto hexagonal Ti ($h$-Ti) surface planes of monolayer $Ti_3C_2$, which serve as the growth substrate. Combined with density functional theory (DFT), the homoepitaxial growth of single layer $h$-TiC is explained as a delicate interplay between energy barriers[24,25], i.e., low diffusion barrier of Ti and C adatom on $h$-Ti surface, high surface energy of $h$-Ti surface, high step-edge energy, and high binding energy of $h$-TiC adlayer. We employed ReaxFF-based hybrid force biased Monte Carlo (fbMC)/molecular dynamics (MD) simulations to further demonstrate these growth dynamics.

## Results and Discussion

**In situ homoepitaxial growth.** The homoepitaxial growth process is experimentally investigated using a Nion UltraSTEM 100 microscope at a vacuum of roughly $10^{-9}$ Torr combined with a Protochips Fusion in situ heating system (see Methods). Monolayer $Ti_3C_2$ 2D material is prepared from annealing monolayer $Ti_3C_2T_x$ flakes inside the microscope at above 500 °C (Fig. 1a). $Ti_3C_2T_x$ is the chemical exfoliation product from the selective etching of "Al" out of the MAX phase $Ti_3AlC_2$[4,26,27], and $T_x$ denotes functional groups such as –F, –OH and –O. The crystal structure of $Ti_3C_2T_x$ can be regarded as a Ti–C–Ti–C–Ti quintuple layered-structured $Ti_3C_2$ core terminated by $T_x$ on two surfaces (Fig. 1b). When projected along the $c$ axis, the three Ti layers labeled as CAB form a hexagonal pattern of three different sites (Fig. 1b), which is also displayed in the corresponding STEM image acquired at room temperature (Fig. 1c). The bright dots with similar intensity in STEM images represent projected Ti atoms from the three different layers, while the C atoms cannot be detected due to their overlap with heavier Ti atoms along the $c$

axis[28]. The dark contrast (indicated by white dotted circles) results from Ti vacancies on the bottom and top surfaces from the etching process[27,29]. At room temperature, due to the surface stabilization by functional groups and the much lower thermal energy, the $Ti_3C_2T_x$ flakes are stable under e⁻ beam irradiation (Supplementary Movie 1).

At 500 °C, monolayer $Ti_3C_2$ flakes are formed following the removal of $T_x$ from $Ti_3C_2T_x$ MXene, as evidenced by the disappearance of $T_x$-associated oxygen K edge in the electron energy loss spectrum (EELS) acquired after heating and beam irradiation (Fig. 1f). Additionally, $Ti_3C_2$ flakes undergo significant morphology changes after electron-beam irradiation and heating for about 20 min (see Fig. 1d and Supplementary Movie 2). The most prominent features are: (1) dark-contrast faceted pores with edges aligned along {100} planes, and (2) bright-contrast triangular islands framed by black and blue dashed triangles that suggest a growth process of an additional atomic layer at least consisting Ti atoms. As revealed by EELS elemental mapping, the adlayer areas have higher relative C concentration than $Ti_3C_2$ (Supplementary Fig. 1). The EELS result thus favors a hexagonal TiC adlayer structure rather than pure Ti adlayer structure. Moreover, DFT suggests pure Ti adatoms tend to form 3D clusters with lower formation energy than 2D Ti adlayer when C atoms are absent.

To solve the crystal structure of the triangular islands, we first consider the stable crystal structure of a single $h$-TiC adlayer on a $Ti_3C_2$ substrate (Fig. 1g). Eight models are constructed based on (1) different Ti stacking sequences, CABC (I, IV, VI) or CABA (II, III, V) or CABB (VII, VIII); (2) with C occupying octahedral interstitial sites (I, II) or tetrahedral interstitial sites (III–VIII) or surface tetrahedral sites (V–VIII) (Fig. 1g). DFT total energy calculation predicts the CABC stacking of Ti layers with C atoms at octahedral interstitial sites (configuration I) to be the ground state of an $h$-TiC adlayer on $Ti_3C_2$. When more $h$-TiC adlayers are grown, DFT predicts BCABC and ABCABC stacking for two and three $h$-TiC adlayers on $Ti_3C_2$, respectively (Supplementary Fig. 2). In general, the local crystal structure of nanoscale as-grown areas is modified to $Ti_{n+3}C_{n+2}$ if $n$ layers of $h$-TiC adlayers grow on the $Ti_3C_2$ substrate.

Theoretical STEM images simulated using DFT-optimized crystal structure of $Ti_{n+3}C_{n+2}$ exhibit unique patterns that can readily be used to unambiguously identify the number of $h$-TiC adlayers solely from the experimental STEM images (Fig. 1h). All the triangular areas in Fig. 1e are identified as one $h$-TiC adlayer with a local structure of $Ti_4C_3$ (one Ti site is brighter than the other two) or two $h$-TiC adlayers with a local structure of $Ti_5C_4$ (two Ti sites are brighter than the third). Considering that both the top and bottom $h$-Ti surfaces of a suspended $Ti_3C_2$ flake can function as substrates, the growth is mostly limited to one $h$-TiC adlayer on each $h$-Ti surface, while $Ti_5C_4$ areas result from overlapping of single $h$-TiC adlayers grown on both surfaces. The islands are generally several nanometers and their size is likely limited by the available source of Ti and C atoms. Future growth of large area TMCs will need external metal and carbon sources beyond the substrate material.

Heating at 1000 °C significantly accelerates the homoepitaxial growth process. With the electron beam blanked, the sample was kept at 1000 °C for several seconds and then cooled down to room temperature. Solely activated by thermal energy, $h$-TiC adlayers rapidly grow on both $h$-Ti surfaces of monolayer $Ti_3C_2$ flakes, forming network of $Ti_5C_4$ and $Ti_4C_3$ regions (Fig. 1e) that are much larger than those formed at 500 °C. The $h$-TiC adlayers exhibit irregular shapes, possibly due to the non-equilibrium nature of fast diffusion and growth at the elevated temperature. Growth of mostly single $h$-TiC adlayers on $Ti_3C_2$, strongly suggests an island type FM growth mode that prefers 2D layer

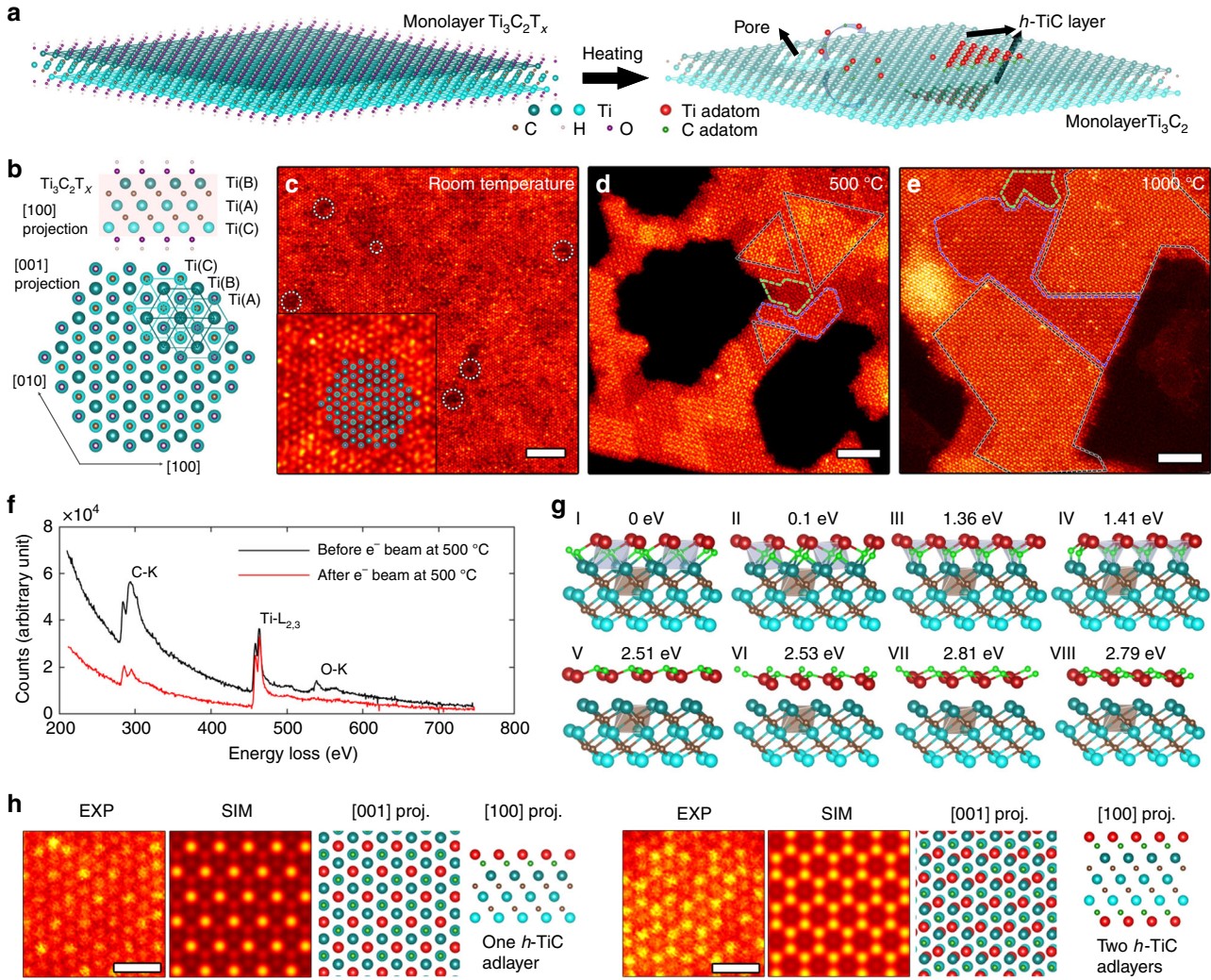

**Fig. 1** Homoepitaxial growth of MXene. **a** Schematic of the homoepitaxial growth process to form single layer h-TiC on a monolayer $Ti_3C_2$ substrate through electron-beam irradiation and heating. Ti atoms from different layers are plotted using different shades of blue. **b** Crystal structure of monolayer $Ti_3C_2T_x$ viewed from [100] and [001] zone axes. **c** Atomic resolution STEM image acquired from monolayer $Ti_3C_2T_x$ along the [001] zone axis at room temperature. Areas of Ti vacancies are indicated by white dotted circles. Inset shows comparison between atomic resolution STEM image and projected crystal structure of $Ti_3C_2T_x$. **d** A STEM image acquired after heating and electron-beam irradiating monolayer $Ti_3C_2$ flakes inside the microscope at 500 °C. The black areas are pores, while areas framed by green, blue, and black dashed lines are the substrate ($Ti_3C_2$), nanoscale areas of single h-TiC adlayer on one surface ($Ti_4C_3$), and nanoscale areas of single h-TiC adlayers on both surfaces ($Ti_5C_4$), respectively. **e** A STEM image acquired after heating MXene flakes inside the microscope at 1000 °C. **f** EEL spectra before and after e-beam irradiation at 500 °C for the C-K edge at 284 eV, Ti-$L_{2,3}$ edge at 456 eV, and O-K edge at 532 eV. **g** Perspective view of eight different stacking configurations (I-VIII) of h-TiC adlayer on a $Ti_3C_2$ substrate. The relative total energy (eV) to the ground state (I) is shown for each structure. **h** Experimental (EXP) and simulated (SIM) STEM images of $Ti_4C_3$ and $Ti_5C_4$, and the corresponding crystal structures projected along [001] and [100] zone axes. The scale bars in **c**–**e** are 2 nm, while the scale bars in **h** are 0.5 nm

formation. Moreover, the homoepitaxial growth here leads to the formation of a mixture of new MXenes $Ti_4C_3$, $Ti_5C_4$, which are predicted to possess excellent energy storage properties (Supplementary Fig. 3)[30], but have never been synthesized using conventional top-down exfoliation methods. To elucidate the growth mechanism, we employ time-dependent STEM imaging and DFT.

**Growth mechanisms and kinetics.** Activated by electron-beam irradiation and/or thermal exposure, Ti and C atoms in monolayer $Ti_3C_2$ flakes migrate onto the h-Ti surface, providing Ti and C adatoms as the source material for growth, thereby leaving pores in the substrate. This is the first step of homoepitaxial growth. At 500 °C, individual STEM image frames (Fig. 2a–i) are acquired at different times to show pore evolution from a three-

Ti-vacancy (3-$V_{Ti}$) cluster as indicated by dashed white circles in Fig. 2a. The 3-$V_{Ti}$ cluster is likely formed by the removal of three adjacent Ti atoms from one of the surface layers during chemical etching[27]. The Ti vacancy cluster grows as Ti atoms migrate onto the surface (Fig. 2b, c). The remaining Ti atoms in the center rearrange to adopt the minimum energy state and eventually there are only enough Ti atoms to form a single Ti layer in the center (Fig. 2d). From Fig. 2d–f, the central Ti atom (indicated by the green circle) finally diffuses away. The Ti single layer area immediately collapses, and then a vacuum pore continues to expand as shown from Fig. 2g–i.

The energy barriers associated with migration of C and Ti atoms onto the h-Ti surfaces are investigated using DFT (Fig. 3a–e). In defect-free $Ti_3C_2$, the energy barriers for a C atom (green path from "I" to "F" in Fig. 3a) or a Ti atom (red path from "I" to "F" in Fig. 3c) in the surface layers to move onto

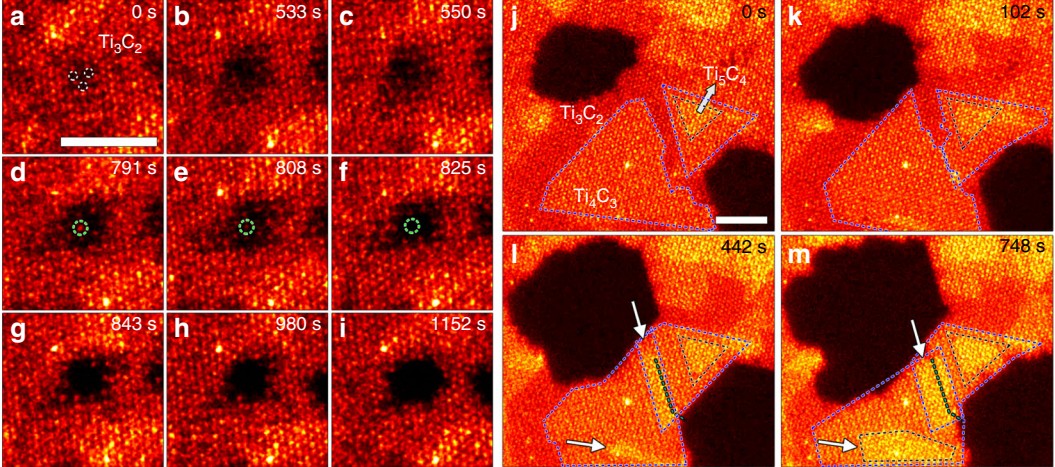

**Fig. 2** Dynamic growth process as revealed by atomic resolution STEM imaging. **a–i** Pore expansion captured by a series of time-lapsed STEM images acquired from a monolayer $Ti_3C_2$ area following electron-beam irradiation and heating at 500 °C. The initial $3\text{-}V_{Ti}$ cluster is indicated by three white dotted circles. The last Ti atom that migrates to the surface before pore formation is indicated by the green dotted circle. **j–m** Time-lapsed STEM images showing the dynamic expansion of island growth of $h\text{-}TiC$ adlayers ($Ti_4C_3$: blue dotted lines, $Ti_5C_4$: black dotted lines) on $Ti_3C_2$ substrate at 500 °C. White arrows in **l** and **m** indicate overlap of $h\text{-}TiC$ adlayers on bottom and up surfaces to form $Ti_5C_4$. Cyan dashed lines in **l** and **m** indicate the projected grain boundary where the two adlayers first encounter in **k**. The scale bars are 2 nm

the surface are 2.91 eV and 5.32 eV, respectively. Both energy barriers are quite high, especially for the Ti atom migration that requires breaking of strong Ti–C bonds and overcoming repulsive force from other Ti cations. The migration barrier, however, is reduced by the presence of defects. Similar to the experimental condition in Fig. 2a, when a $3\text{-}V_{Ti}$ cluster (white balls in Fig. 3b, d) is present, the green C atom at position "I" in Fig. 3b is only bonded to three Ti atoms in the middle layer instead of 6 Ti atoms for defect-free $Ti_3C_2$. Consequently, a migrating C atom only needs to break 3 Ti–C bonds and the migration barrier is reduced to 0.97 eV (path $3\text{-}V_{Ti}$, Fig. 3b). Although a $3\text{-}V_{Ti}$ cluster does not help migration of Ti atoms (Fig. 3d), the migration barrier of a Ti atom near a C vacancy ($V_C$) is reduced to 2.56 eV (path $V_C$ in Fig. 3e) because it possesses one fewer Ti–C bond. Other possible migration paths for surface Ti and C atoms at various defective environments show comparable migration energy barriers (Supplementary Figs. 4–6). For Ti atoms in the middle layer, the migration barrier is 6 eV, which can be lowered when all the surrounding Ti and C atoms are moved away (Fig. 2d, Supplementary Fig. 7). In that case, the Ti atom at the middle layer first diffuses to one of the two outer layers, and then migrates to the surface with a barrier of 2.99 eV. The DFT results agree well with experiment, where the migration starts from the defective surface layers, followed by the middle layer. DFT also indicates that C atoms move to the surface first, followed by Ti atoms. The maximum energy transferred to a Ti atom from 100 kV electron beam is 5 eV based on Eq. (1) in Ref. [31] and is sufficient to activate the migration process.

The second step in the growth process is island formation of $h\text{-}TiC$ adlayer on $h\text{-}Ti$ surface, which occurs by the self-assembly of C and Ti adatoms that have already migrated onto the surface. The diffusion barrier of a Ti and a C adatom on the bare surface of $Ti_3C_2$ is 0.10 and 1.04 eV, respectively (Fig. 3f, i), which suggests that Ti atoms are more mobile on the surface as compared to C atoms. When a Ti atom diffuses within proximity of a C atom, it is energetically more favorable for them to bond into a TiC dimer than for them to remain separate. With a diffusion barrier (0.72 eV) lower than a C atom or a Ti–C–Ti trimer (1.24 eV, see Supplementary Fig. 8 for diffusion path and energy for dimers and trimers), the TiC dimer should be a common molecule moving on the surface. When several TiC

dimers cluster together and become immobile because of increased diffusion barriers, they could serve as a nucleation site for bonding of additional TiC dimers, resulting in island formation and expansion. A growth mechanism, assuming that a layer of C atoms grows on the substrate first followed by a layer of Ti atoms, is unlikely considering the very high formation energy of such MXene-like C layers (2.65 eV/atom, see Supplementary Fig. 9 for more details).

An example of island growth is shown in Fig. 2j–m (see Supplementary Movie 2 for more details). Two pores of dark contrast continuously expand from Fig. 2j–m and serve as the Ti and C sources that sustain the growth process. The two islands of single $h\text{-}TiC$ adlayer framed by blue dotted lines are initially separated (Fig. 2j), and they continue to grow strictly limited to lateral expansion to meet and overlap as shown in Fig. 2k–m, respectively. The two adlayers are most likely formed on the top and bottom surfaces, respectively, because their triangular outlines show opposite orientations, which is to be expected assuming the adlayers grow following "CABC" stacking sequence and have the same edge structure (see Supplementary Fig. 10 for more details). This is further confirmed by the fact that when the two $h\text{-}TiC$ adlayers starts to overlap in STEM images to form local $Ti_5C_4$ structure (marked by white arrows in Fig. 2l), the growth is not interrupted, and both grains grow past the initial grain boundary line as indicated by cyan dashed lines. Therefore, previously observed $Ti_5C_4$ regions should mostly result from overlapping of single $h\text{-}TiC$ adlayers growing on both $h\text{-}Ti$ surfaces. It is also possible that the $Ti_5C_4$ region could result from two adlayers growing on one surface. The $Ti_5C_4$ region framed by the black dashed lines (Fig. 2j–m) has edges parallel to the edges of the outer $Ti_4C_3$ area framed by blue dashed lines, suggesting that the two adlayers might be on the same surface. As $Ti_6C_5$ areas are rarely observed and $Ti_7C_6$ were never observed, growth of two adlayers on one surface happens less frequently than growth of a single adlayer on each surface.

When interacting with $Ti_3C_2$ substrate, the 2D $h\text{-}TiC$ adlayer has the lowest formation energy compared to different configurations of cubic TiC nanoparticles (NPs) (Fig. 4a), despite cubic TiC being the most stable titanium carbide structure. The total energy mainly depends on the strong covalent bonds between C atoms in the adlayer and Ti atoms in the surface. The 2D $h\text{-}TiC$

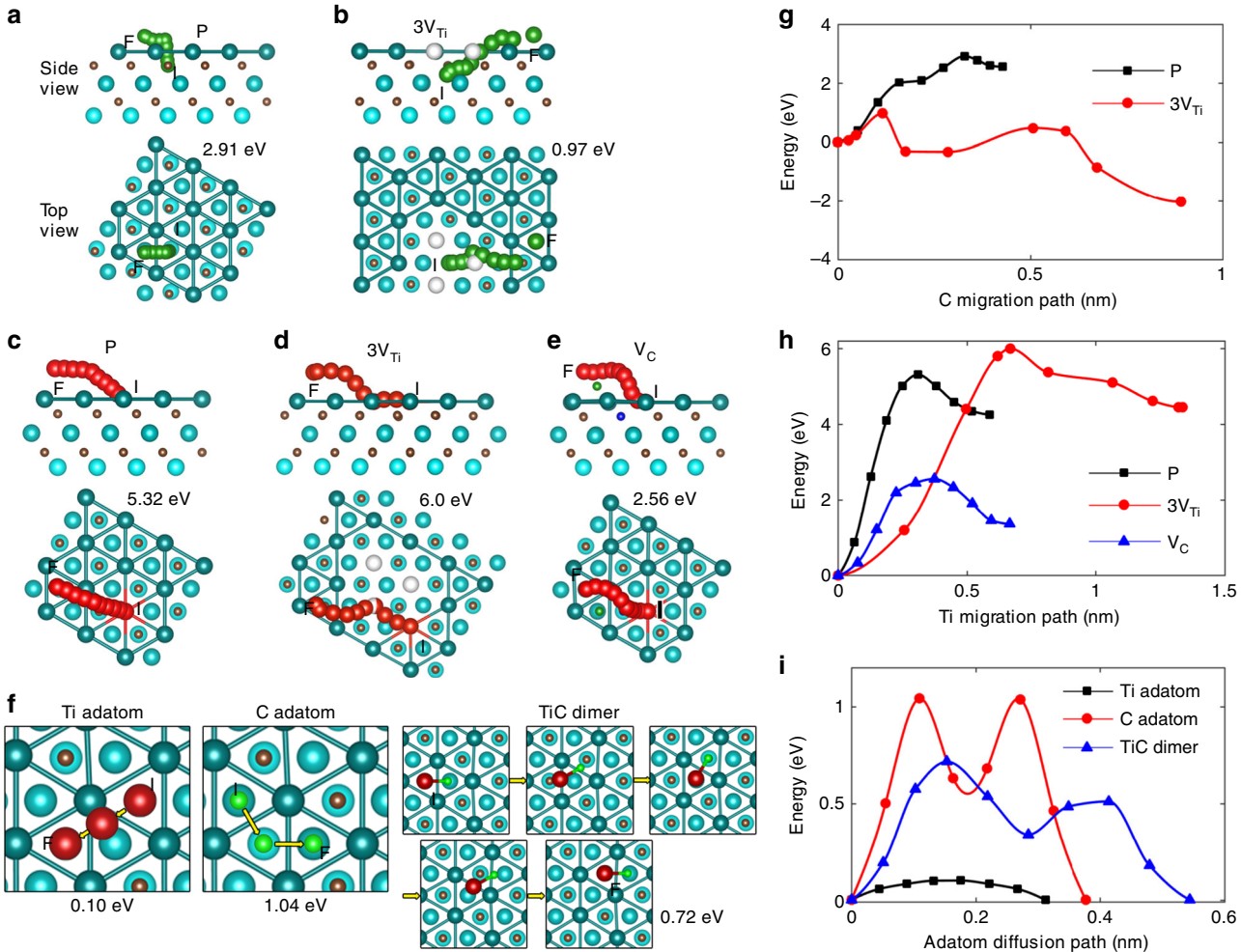

**Fig. 3** Migration and diffusion barriers for Ti and C atoms. **a–e** Carbon atom migration pathways for a perfect crystal ("P", **a**), and with help of 3-$V_{Ti}$ cluster ("3$V_{Ti}$", **b**); and titanium atom migration pathways for a perfect crystal ("P", **c**), with help of 3-$V_{Ti}$ cluster ("3$V_{Ti}$", **d**), and with help of a single C vacancy ("$V_C$", **e**). "I" and "F" denote the initial and final positions of the migration paths. Cyan, Ti; gold, C; red, diffusing Ti atom; green, diffusing C atom; white, Ti vacancy site; blue, C vacancy site. **f** Diffusion pathways of Ti adatom, C adatom, and TiC dimer on the Ti$_3$C$_2$ surface. The migration or diffusion energy barrier for each case in **a–f** is labeled in eV. **g**, **h** Calculated energy along the migration paths for C (**g**) and Ti (**h**) from Ti$_3$C$_2$ body to Ti$_3$C$_2$ surface. **i** Energy along the diffusion paths on the Ti$_3$C$_2$ surface for a Ti adatom, a C adatom, and a TiC dimer

has the largest interaction area with the *h*-Ti surface, the largest number of Ti–C bonds, and therefore the lowest total energy of 0.44 eV/TiC (Fig. 4a). Although a planar TiC has similar contact area as a 2D *h*-TiC layer, the severe mismatch between hexagonal Ti$_3$C$_2$ and tetragonal TiC lattice leads to higher formation energy (0.96 eV/TiC). In contrast, low interacting areas between various cuboid TiC NPs and Ti$_3$C$_2$ substrate lead to high formation energies (1.11 and 1.54 eV/TiC for two different cuboid shapes in Fig. 4a). More configurations and the formation energies can be found in Supplementary Table 1 and Fig. 11. The *h*-Ti surfaces of monolayer Ti$_3$C$_2$ thus serve as an ideal substrate that ensures that growth of a *h*-TiC single layer is energetically favorable, hinting that with a careful selection of the substrate, bottom-up synthesis of TMCs may become a reality by the mechanisms shown by STEM.

**Edge structure and equilibrium shape.** The edge structure of 2D materials critically determines the probability of atom attachment/detachment to/from the edges which impacts growth of 2D materials and tailors the final shape of 2D materials[32–35]. STEM images indicate that *h*-TiC adlayers are mostly terminated with zigzag (ZZ) edges (see Figs. 1c–d, 2j–m, 4d). To help understand

the arrangement of C atoms that are indeterminate in the STEM images, we performed DFT calculations for seven hypothetical edges including four ZZ type edges (ZZ-Ti, ZZ-C, ZZ-Ti + C, and ZZ-C + Ti), and three armchair (AC) type edges (AC, AC + Ti, AC + C) (Fig. 4b). The formation energies ($\gamma$) of different edges depend on Ti chemical potential difference ($\Delta\mu_{Ti}$, defined as $\mu_{Ti}$ - $\mu_{Ti\_bulk}$), as plotted in Fig. 4c (see Supplementary Figs. 12 and 13 for a broader $\Delta\mu_{Ti}$ range). Near the Ti-rich condition ($-0.11 < \Delta\mu_{Ti} < 0$ eV), ZZ-Ti is the most stable edge structure, while ZZ-C becomes the most stable edge structure in a wide range of $-1.50 < \Delta\mu_{Ti} < -0.11$ eV. In both cases, ZZ-type edges are the most stable, in agreement with experimental observation.

Using a Wulff construction based on edge formation energies, the thermodynamic equilibrium shape of 2D *h*-TiC adlayer on Ti$_3$C$_2$ are visualized for $\Delta\mu_{Ti} = -1.54$ eV (Fig. 4e), $-0.50$ eV (Fig. 4f), and 0 eV (Fig. 4g). Generally, edges with lower formation energy are retained during the growth[32,33,36]. Under C-rich conditions ($\Delta\mu_{Ti} = -1.54$ eV), *h*-TiC adlayer adopts the shape of a perfect triangle terminated by ZZ-C edges. At $\Delta\mu_{Ti} = -0.5$ eV, ZZ-Ti edges start to emerge at the triangle vertices. Under Ti-rich conditions ($\Delta\mu_{Ti} = 0$), the equilibrium shape turns into a dodecagon consisting of ZZ-C, ZZ-Ti, and AC edges that

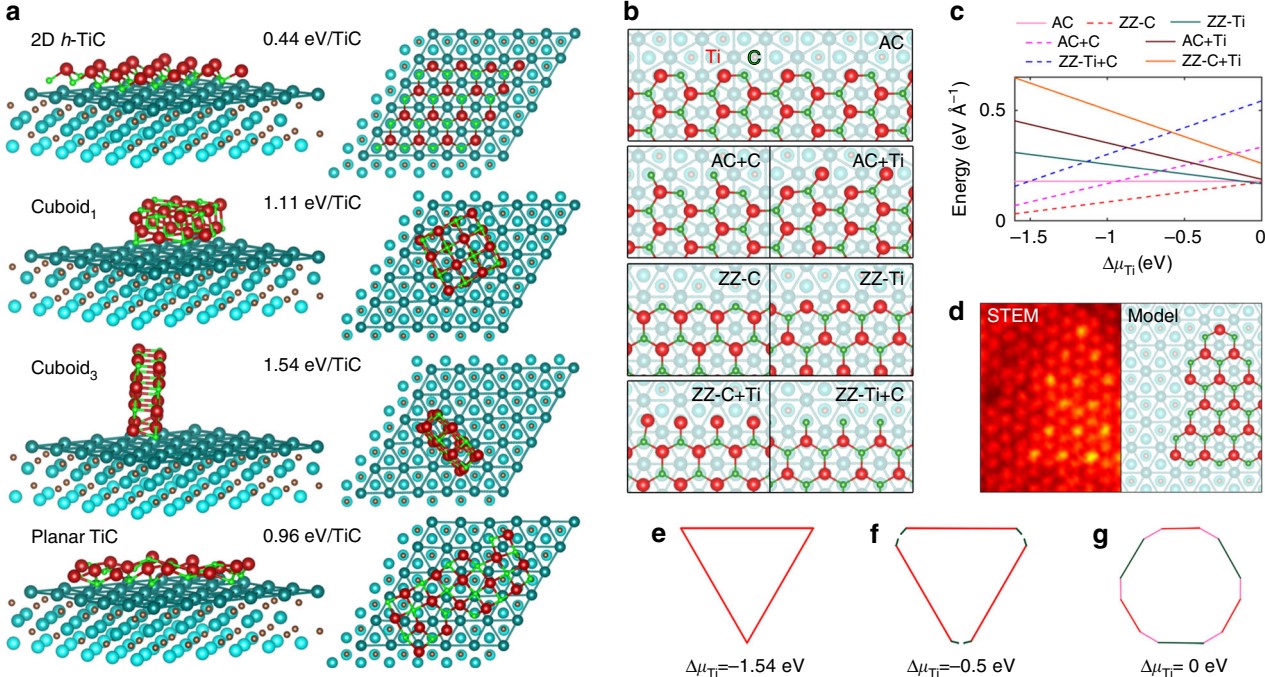

**Fig. 4** Edges and equilibrium shapes of 2D $h$-TiC adlayer on $Ti_3C_2$ surface. **a** Side view (left) and top view (right) of optimized crystal structure and formation energy of an $h$-TiC adlayer, two different cuboid shapes (cuboid$_1$ and cuboid$_3$), and planar TiC, all composed of 16 atoms, on $Ti_3C_2$ surface. **b** Structural models of the edge structure of $h$-TiC adlayer on $Ti_3C_2$, including standard armchair (AC), Ti- and C-oriented zigzag (ZZ-Ti and ZZ-C) edges, and cases when they are terminated with Ti or C atoms. The Ti and C atoms in the $h$-TiC adlayer are red and green respectively. **c** Calculated edge energies of $h$-TiC on $Ti_3C_2$ as a function of Ti chemical potential difference, $\Delta\mu_{Ti}$. **d** An experimental STEM image and simulated model showing the triangular island and the ZZ-C edge crystal structure model. **e–g** The equilibrium shape of $h$-TiC on $Ti_3C_2$ at varying $\Delta\mu_{Ti}$: −1.54 eV (**e**), −0.50 eV (**f**), and 0 eV (**g**). The colored outlines represent different types of edge structure

have similar formation energies. Compared with the experimentally observed adlayers that are typically triangular, it is concluded that experimental growth conditions are C-rich, and the edges are most likely ZZ-C. In Fig. 4d, an experimental triangular adlayer from 500 °C heating shows excellent agreement with the ZZ-C terminated structure model. At 1000 °C (Fig. 1e), $h$-TiC adlayers exhibit more diverse shapes because (1) thermodynamic equilibrium states cannot be reached within the short evolution time, and (2) merging of fast-growing islands obscures the original shapes.

With the knowledge of the $h$-TiC adlayer edge structure, we can now calculate the step-edge barrier for adatoms on $Ti_3C_2$ surface to climb up onto $h$-TiC adlayer through aZZ-C edge. The step-edge barriers for a Ti adatom, a C adatom, and a TiC dimer are 2.23, 1.36, and 2.56 eV, respectively (Supplementary Fig. 14), which are comparable to Ti and C migration energy from the body onto the $Ti_3C_2$ surface (Fig. 3g, h), but much higher than the diffusion barriers of adatoms on the $Ti_3C_2$ surface (Fig. 3i). Energetically, the adatoms prefer diffusing on the $h$-Ti surface until attaching on a $h$-TiC adlayer, to climbing up onto a $h$-TiC adlayer. The vertical growth of $h$-TiC adlayer would only occur after the $Ti_3C_2$ surface is fully covered by the first $h$-TiC adlayer, which is difficult to achieve experimentally. In rare cases (see the brightest region in Fig. 1e, and black dashed triangle in Fig. 2j–m), localized growth of multiple $h$-TiC adlayer is possible when TiC dimers cannot diffuse out.

**ReaxFF simulations of homoepitaxial growth**. The dynamic growth process is simulated using the reactive force field (ReaxFF) method, which provides atomistic level details as well as a realistic representation of chemistry[37–39], at a larger length scale than DFT. A ReaxFF force field that was developed for $Ti_3C_2$

MXene structures[40] is employed here, and is shown to correctly predict energy barrier for Ti and C adatoms diffusion on $Ti_3C_2$ substrate (Supplementary Fig. 15). In the simulations, two initial pores were introduced by deleting Ti and C atoms and placing them randomly at the surface a monolayer $Ti_3C_2$ flake consisting of around 10,000 atoms and a size of 112 by 129 Å. Also, a small seed cluster of Ti and C adatoms was created near the pores. Clustering of surface adatoms into crystalline triangular islands (indicated by white dashed lines in Fig. 5d) is observed by using hybrid force-bias Monte Carlo (fbMC) and MD simulation at 1500 K. Figure 5a–d focuses on the island growth dynamics, showing the formation of a triangular shaped $h$-TiC adlayer from scattered Ti and C atoms, which is analog to the growth process of MBE. It should be noted here that because of time-scale limitation of these simulations, we chose higher temperatures than experiment to accelerate the dynamics of the system during a short time scale. For simulating pore expansion, temperature was set to 2500 K to overcome higher energy barriers of Ti and C diffusion to the surface. As shown in Fig. 5e–h, the initial pores expand during the simulation and the atoms migrate to the MXene surface. As revealed in ReaxFF simulation, diffusion of Ti and C adatoms leads to the nucleation of the local $h$-TiC island, which then expand as more Ti and C atoms are attached to the edges, similar to the lateral growth of islands in Fig. 2j–m.

In summary, by combining experiments (in situ STEM) with theory (DFT) and simulations (ReaxFF), the bottom-up homoepitaxial FM growth of 2D $h$-TiC is demonstrated and is attributed to the intricate interaction between the $Ti_3C_2$ substrate and the $h$-TiC adlayer, and the unique growth mechanism that the source atoms are provided completely through surface diffusion on the substrate. Combined with low diffusion barrier and high step-edge barrier, such growth has the advantage of

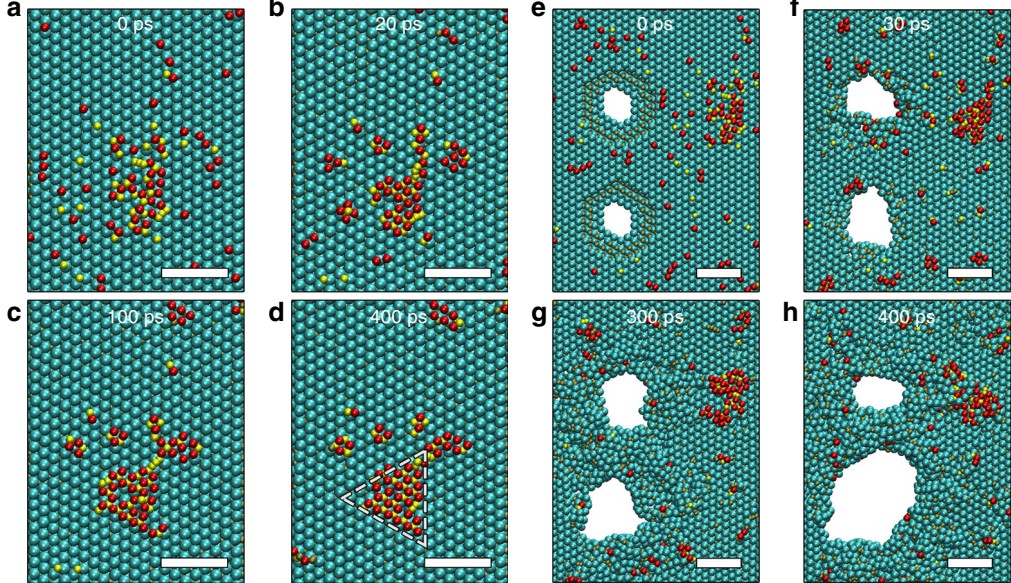

**Fig. 5** ReaxFF Molecular dynamic simulation showing the structural evolution in MXene. **a–d** Island growth observed for an initial configuration with a Ti–C seed after different time periods of ReaxFF force-bias Monte Carlo/molecular dynamics (fbMC/MD) simulation at 1500 K. **e–h** Pore expansion observed for initial configuration with two pores and Ti–C seed after different time of fbMC/MD simulation at 2500 K. The scale bars are 1 nm. (adatom Ti: red, Ti: cyan, C: gold, adatom C: yellow)

effectively confining source atoms on the substrate, leading to site-specific and localized 2D TMC growth which is otherwise difficult to achieve. The successful growth on ultrathin and *h*-Ti terminated $Ti_3C_2$ substrates demonstrated here may lead to search for alternative substrates that promote atomic layer growth. For example, the (0001) surfaces in hexagonal close-packed (hcp) metal compounds (WC) or pure hcp metal (Sc, Ti, Zr, or Hf) with a hexagonal metal (*h*-M) surface layer could be used to grow TMC, similar to CVD growth of hcp β-$Nb_2N$ and γ-$Ta_2N$ on Nb and Ta thin film[41]. Transfer after growth might be achieved by etching out the metallic substrate[9]. Therefore, implementation of such concept at a larger scale and using a source that can provide continuous flux of source atoms through surface diffusion on an atomic-flat substrate with long diffusion length for the source atom species, would lead to large-scale bottom-up synthesis methods for 2D TMC.

## Methods

**Sample preparation**. $Ti_3C_2T_x$ was synthesized using the minimally intensive layer delamination (MILD) method as previously reported[26]. To prepare $Ti_3C_2T_x$-MXene with minimal defects, the etchant solution was prepared by completely dissolving 0.4 g LiF in 10 ml of 6 M HCl in a 50 ml-polypropylene plastic vial after which 0.4 g of $Ti_3AlC_2$ was gradually added to the etchant solution and the reaction allowed to proceed for 24 h at 35 °C. The acidic product was transferred to a 150-ml centrifuge tube and washed three times (5 min each cycle) with deionized water (DI $H_2O$) via centrifuging at 3500 rpm. The sediment was transferred to 50-ml centerfuge tube and diluted with 40 ml DI $H_2O$ and manually shaken for a couple of minutes before it was centrifuged again at 3500 rpm for 1 h. The collected dark green supernatant was a 1.5 mg/ml of $Ti_3C_2T_x$ colloidal solution.

**STEM characterization and simulation**. $Ti_3C_2T_x$ samples were prepared by drop-casting $Ti_3C_2T_x$ colloidal solution onto a commercial microelectromechanical systems (MEMS) based in situ heating platform from Protochips, Inc. with C or SiN membrane. For the SiN membrane, arrays of holes were created using a $Ga^+$ ion focused ion beam to ensure that freestanding $Ti_3C_2T_x$ flakes do not interact with the substrate during heating. In situ STEM heating experiments were performed using a Nion UltraSTEM, operating at 100 kV with a beam current of 40 pA and equipped with a spherical aberration ($C_s$) corrector to achieve the 1 Å spatial resolution. A convergence angle of 31 mrad was used, with HAADF detector inner and outer collection angles of 86 mrad and 200 mrad, respectively. HAADF-STEM image simulation was performed using the code from Kirkland[42].

**Density functional theory simulations**. The total energy first-principles calculations were performed using DFT within the local density approximation (LDA) and the projector-augmented wave (PAW) method[48], as implemented in Vienna Ab-initio Simulation Package (VASP)[49]. For the exchange-correlation energy, we used the Perdew–Burke–Ernzerhof (PBE) version of the generalized gradient approximation (GGA)[50]. A plane-wave cutoff energy of 500 eV was sufficient to ensure convergence of the total energies to 1 meV per primitive cell. The underlying structural optimizations were performed using the conjugate gradient method, and the convergence criterion was set to $10^{-5}$ eV/cell in energy and 0.05 eV/Å in force. A large 6 × 6 and 8 × 8 monolayer $Ti_3C_2$ supercell has been used to calculate the migration barriers and edge formation energies of hexagonal 2D TiC layer. To determine the energy barriers and minimum energy paths of proposed diffusions, we used the climbing image nudged elastic band method (CI-NEB) implemented in VASP[51]. Six to eight images were simulated between initial and final states. The NEB path was first constructed by linear interpolation of the atomic coordinates and then relaxed until the forces on all atoms were < 0.05 eV/Å. The formation energy or the adsorption energy is calculated as

$$E = \frac{E_{ad+Ti_3C_2} - E_{Ti_3C_2} - nE_{ad\_bulk}}{n}$$

**Edge formation energy and Wulff construction**. The formation energy of armchair edge ($\gamma_{AC}$) can be directly obtained based on AC nanoribbon (NR) as

$$\gamma_{AC} = \frac{E_{TiC+Ti_3C_2} - E_{Ti_3C_2} - n_{TiC}\mu_{TiC}}{2L} \quad (1)$$

The formation energy of ZZ-C and ZZ-Ti edges is defined as

$$\gamma_{ZZ-C} = \frac{E_{TiC+Ti_3C_2} - E_{Ti_3C_2} - n_{TiC}\mu_{TiC} - n_C\mu_C - \mu_{Ti}}{3L} \quad (2)$$

$$\gamma_{ZZ-Ti} = \frac{E_{TiC+Ti_3C_2} - E_{Ti_3C_2} - n_{TiC}\mu_{TiC} - n_{Ti}\mu_{Ti} - \mu_C}{3L} \quad (3)$$

In eqns (1)–(3), $E_{TiC+Ti_3C_2}$ and $E_{Ti_3C_2}$ are the calculated total energies of *h*-TiC flakes/NRs on $Ti_3C_2$ surface and the $Ti_3C_2$ monolayer, respectively. $n_{TiC}$ is the number of TiC pairs, $\mu_{TiC}$ is the energy of a TiC pair in *h*-TiC layer as $\mu_{TiC} = E_{Ti_4C_3} - E_{Ti_3C_2}$. $\mu_{Ti}$ and $\mu_C$ are the chemical potential of Ti and C species, respectively. Under thermodynamic equilibrium condition, $\mu_{Ti}$ and $\mu_C$ satisfy $\mu_{TiC} = \mu_{Ti} + \mu_C = \mu_{Ti}^{bulk} + \mu_C^{bulk} + \Delta H_{TiC}$, where $\Delta H_{TiC}$ is the formation enthalpy of *h*-TiC as $\Delta H_{TiC} = E_{Ti_4C_3} - E_{Ti_3C_2} - E_{Ti} - E_C$. The range of $\mu_{Ti}$ can be deduced as $\mu_{Ti}^{bulk} + \Delta H_{TiC} \leq \mu_{Ti} \leq \mu_{Ti}^{bulk}$, where the upper (lower) limit corresponds to an Ti-rich (C-rich) condition and $\mu_{Ti}$ ($\mu_C$) is given the total energy of the bulk Ti (C). $n_C$ ($n_{Ti}$) is the number of extra C (Ti) atoms in the triangular TiC flakes, $L$ is the

length of the TiC edge in the unit of angstrom, and the factor 2 or 3 accounts for the number of identical edges in each model considered, 2 for NR models and 3 for triangular domain models.

For an arbitrary chiral edge with chiral angle $\chi$, it contains ZZ and AC sites. Thus, the formation energies of a chiral edge ($\gamma(\chi)$) can be obtained using the following expression:

$$\gamma(\chi) = |\gamma_0| \cos(\chi + C)$$

where

$$|\gamma_0| = 2\sqrt{\left(\gamma_{AC}^2 + \gamma_{ZZ-X}^2 - \sqrt{3}\gamma_{AC}\gamma_{ZZ-X}\right)}$$

$$C = \arctan\frac{\sqrt{3}\gamma_{AC} - 2\gamma_{ZZ-X}}{\gamma_{AC}}$$

with the subscript $X = C$ at $-30° < \chi < 0°$ or $X = Ti$ at $0° < \chi < 30°$.

**ReaxFF simulations**. ReaxFF-based hybrid fbMC/MD[43] were used to model island growth and pore expansion on MXene structure. ReaxFF is a bond-order-based reactive force field technique which can consider bond formation and bond dissociation during atomistic simulation. The total energy in ReaxFF includes bond-order dependent energy terms such as bond, angle, and torsion, and non-bonded interaction terms including Coulomb and van der Waals interactions. For the Coulomb interaction, ReaxFF uses the electronegativity equalization method (EEM) to determine the atomic charges. MD is a robust atomistic technique for material modeling providing information about dynamical behavior of the system. However, for some long-term equilibrium processes such as phase transition and growth, a method faster than MD is desirable. An alternative is to move all atoms with greater probability in the direction of instantaneous force by using the method called force-bias monte carlo (fbMC)[44,45]. Implementation of fbMC with ReaxFF has been successfully performed previously for carbon nanotube growth in Ni-clusters[46]. Here, we have combined both MD and fbMC to keep the trajectories and dynamics of the system by using MD and accelerate the process by using fbMC. In this technique fbMC and MD schemes are applied to the system alternatingly. All simulations have been performed with the ADF[47] package.

**Data availability**. The data that support the findings of this study are available from the corresponding authors upon request.

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

## Acknowledgements

Research was supported as part of the Fluid Interface Reactions, Structures and Transport (FIRST) Center, an Energy Frontier Research Center funded by the U.S. Department of Energy, Office of Science, Office of Basic Energy Sciences. In situ aberration-corrected STEM imaging was conducted at Oak Ridge National Laboratory's Center for Nanophase Materials Sciences (CNMS), a U.S. Department of Energy Office of Science User Facility. This research used resources of the National Energy Research Scientific Computing Center, a DOE Office of Science User Facility supported by the Office of Science of the U. S. Department of Energy under Contract No. DE-AC02-05CH11231.

## Author contributions

X.S. and R.R.U. conceived the idea, designed and carried out the experiments, and analyzed the data. M.A., B.A., and Y.G. synthesized the MXene samples. X.S. performed in situ STEM experiments. Y.X., W.S., P.R.C.K. performed DFT simulation. D.E.Y., R.L., A.O., A.C.T.D. conducted ReaxFF simulations. X.L. and K.X. helped understand the growth mechanism. X.S. and Y.X. drafted the manuscript. All authors contributed to interpretation of the results and writing the paper. X.S. and Y.X. contributed equally to the paper.

## Additional information

**Competing interests:** The authors declare no competing interests.

