## [Peer Review File · Nature Communications]

Reviewers' Comments:

Reviewer #1:

Remarks to the Author:

The paper "In situ atomistic insight into the growth mechanisms of single layer 2D transition metal carbides (MXenes)" present experimental results on the homoepitaxial growth of TiC on surfaces of Ti₃C₂ MXene substrates. Thermal excitation and electron beam irradiation locally damage the Ti₃C₂ substrates, then the displaced atoms migrate and form atomic layer of h-TiC on top and/or bottom surface of the undamaged area. In fact, there are so many works on electron beam induced epitaxial growth in other material system, such as metal oxide, the major difference between this work and previous works is that the substrate this work adopted is much thinner, the experimental results are predictable. I do not have major objections with respect to the correctness of the work, but I have some other questions, which I believe the authors need to properly address to be considered further.

1. The author declare the homoepitaxial growth of h-TiC is activated by thermal energy and accelerated by e-beam irradiation in L 55, P2, but there is no basis for such a conclusion in the manuscript, maybe the author should do another controlled experiment to identify that the growth can be activated only under thermal excitation.
2. The author assert that the adlayers are h-TiC in L 109, P5, but they did not rule out other possible configurations, such as Ti layers, or layers reconstructed by Ti and functional groups.
3. What's the different between the triangular islands with different orientation angle in HRSTEM images(Fig.1d, Fig.2j-m)?
4. In P9, the author consider that the two triangular islands with different orientation angle grow on top and bottom surfaces respectively, because the growth is not interrupted when two islands overlap. But it cannot conclude that other observed Ti₅C₄ regions also result from overlapping of adlayers growing on both surfaces. The Ti₅C₄ regions in Fig. 2j can result from two layer of TiC on the same surface.
5. This work is so special, it looks like an extreme case of epitaxial growth of TiC on TiC substrate, the only thing is the thickness of the substrate is ultrathin then can be labeled as Ti_{n+1}C_n (n>1); although the author can obtain small scale Ti₄C₃ and Ti₅C₄ because the substrate is Ti₃C₂ and the growth source is limited, it's hard to predict the case in large scale synthesis under continuous flux of source. In my opinion, the significant of this work seems not as big as the author claimed.

Reviewer #2:

Remarks to the Author:

In this study, the authors used in-situ STEM technology combined with DFT and MD simulations to demonstrate a clear strategy for bottom-up homoepitaxial Frank-van der Merwe growth of 2D h-TiC on single layered Ti₃C₂ MXene. This is a piece of nice work with high novelty, in which new phenomenon is observed and the theoretical calculations explain the experimental observations very well. I would like to recommend its publishing in Nature Communications after the following comments are addressed.

- 1) The authors demonstrated that the separated triangle h-TiC layers in Fig. 2j belong to the top and bottom surfaces, respectively. However, the depth of focus in STEM mode is very small and the contrast of two h-TiC layers are normally different at the same defocusing value. STEM simulation would be helpful to confirm this speculation. Besides, it is possible that the two h-TiC layers locate on the same surface because when they meet together, the formed grain boundary (Fig. 2k) could act as the nucleation site, due to its low energy, to grow a new h-TiC layer on their top to form Ti₅C₄ regions.
- 2) Why should the migration paths marked in Fig. 3b and 3f be the only way? Dose a different migration path design influence the calculated energy barriers in the end?
- 3) The authors should give the explanations on the terms "P, -S, -B, V1-F, V2-F, V1-N, V2-N" in Supplementary Figure S5-S7?
- 4) The authors mentioned that Ti_{n+3}C_{n+2} with a "ABCABC..." stacking configuration in analogy to

rock-salt structured TiC is the most stable structure. Is there any relationship between the stability of $Ti_{n+3}C_{n+2}$ and the number of TiC layers?

5) The growth method is a little complex because monolayer Ti_3C_2 substrate with h-Ti surface should be obtained first in order to grow single layer h-TiC. Furthermore, to study the properties and applications, it is necessary to transfer the as-grown samples onto target substrates. Could the authors give some discussions on these two points?

6) About the universality of the growth concept, the authors pointed out that it would lead to bottom-up synthesis methods of various 2D materials beyond 2D TMC. Could the authors give more discussions on the materials that might be synthesized by this method?

Reviewer #3:

Remarks to the Author:

This manuscript reports homoepitaxial growth of atomic layers on MXenes stimulated by electron beams. The detailed growth mechanism of TiC monolayer has been investigated by combining in-situ STEM, DFT calculations, and ReaxFF MD simulations, which synergically offer insights into the growth process. In addition, using the electron beam irradiation, the authors are able to obtain new MXenes that have not been synthesized before. I anticipate that this work will further stimulate investigations on the bottom-up synthesis of MXenes that are highly desired due to their potential applications. I therefore strongly recommend its publication, after the authors address the following issues:

1. Scale bars are missing at several places: Figure 1h and Figure 5.
2. What is the feasible range of chemical potential of Ti (Fig. 4, SI Figure 11-12)?
3. Would the edges of the islands be different from the edges of pores?
4. In the DFT calculations, the authors used the approaches established by Liu and Yakobson et al to study direction-dependent edge energy and growth kinetics. Please credit those papers: PNAS, 2012, 109 (38) 15136-15140, Phys. Rev. Lett. 2010, 105, 235502. Also, recently MXenes have been demonstrated to be promising electrodes for 2D electronics (J. Am. Chem. Soc., 2016, 138 (49), pp 15853–15856), it would be informative to the readers if the authors can discuss this in the introduction.

Response to Reviewers' Comments

We would like to thank the reviewers for their positive comments and useful suggestions.

Reviewer #1

The paper "In situ atomistic insight into the growth mechanisms of single layer 2D transition metal carbides (MXenes)" present experimental results on the homoepitaxial growth of TiC on surfaces of Ti₃C₂ MXene substrates. Thermal excitation and electron beam irradiation locally damage the Ti₃C₂ substrates, then the displaced atoms migrate and form atomic layer of h-TiC on top and/or bottom surface of the undamaged area. In fact, there are so many works on electron beam induced epitaxial growth in other material system, such as metal oxide, the major difference between this work and previous works is that the substrate this work adopted is much thinner, the experimental results are predictable. I do not have major objections with respect to the correctness of the work, but I have some other questions, which I believe the authors need to properly address to be considered further.

R. We would like to thank the reviewer for useful comments to improve the quality of the manuscript.

1. The author declare the homoepitaxial growth of h-TiC is activated by thermal energy and accelerated by e-beam irradiation in L55, P2, but there is no basis for such a conclusion in the manuscript, maybe the author should do another controlled experiment to identify that the growth can be activated only under thermal excitation.

R. We would like to thank the reviewer for pointing out this issue. At 500 °C, the homoepitaxial growth is activated by combined beam irradiation and thermal energy. The figure below shows how the central irradiated area (inside the red circle) shows pore formation and homoepitaxial growth, while the peripheral area (outside the red circle) without beam irradiation is still intact. We attribute such difference to the extra energy provides by the e-beam. It also seems like the e-beam plays an important role in removing the surface contamination and surface function groups. However, at 1000 °C, the growth happened without any beam irradiation. With the beam

blanked, the sample was heated to 1000 °C for several seconds and then cooled down to room temperature. As shown in Fig. 1e, the resulting morphology indicates homoepitaxial growth. Therefore, at higher temperature, such homoepitaxial growth happens only under thermal excitation. We have clarified the activation energy in the manuscript in Line 56, Page 2.

Figure R1. A low-magnification STEM image of a MXene flake after homoepitaxial growth at 500 °C. The area with electron beam influence is roughly indicated by the red dashed oval.

2. The author assert that the adlayers are h -TiC in L 109, P5, but they did not rule out other possible configurations, such as Ti layers, or layers reconstructed by Ti and functional groups.

R. Beside STEM imaging, the formation of the h -TiC is also supported by EELS elemental maps as shown in Supplementary Figure S1. The areas with adlayers tend to have higher relative C concentration, which rules out structure of a pure Ti layer. The functional groups are mostly –OH or –O for the MXene samples used in this study. After heating and beam irradiation, the O content has been significantly reduced as confirmed by EELS in Figure 1f. Therefore, the contribution of the functional groups should be minimal. DFT simulation also shows that the formation energy of a complete Ti layer is 0.333 eV/atom compared to bulk Ti on Ti_3C_2 surface. The formation energy should be even higher for the formation of incomplete Ti layer due to the edges. Thus, without associated C atoms, Ti should be preferred to form 3D Ti cluster, which is not observed in our experiments. Therefore, the adlayer most likely be h -TiC, based on evidence

from both experiment and theory. We have moved the EELS discussion to Line 111, Page 5 and also added discussion on the formation energy of pure Ti adlayer.

3. *What's the different between the triangular islands with different orientation angle in HRSTEM images (Fig.1d, Fig.2j-m)?*

R. We thank the reviewer to notice this very interesting point. The different orientations most likely result from growth on different surfaces. For example, if the top adlayer and the bottom adlayer both have the same edge structures, then they should exhibit inversed orientations (180°) as shown in the figure below. This also provides a convenient way to tell if two adlayers are on different surfaces. We have added description in the manuscript in Line 213 Page 9 and added Supplementary Figure 10 in the SI to clarify such orientation difference. We also modified the schematics in Figure 1a such that the two adlayers on the surface and bottom have different orientations.

Figure R2. Orientation difference between triangular adlayers on the top surface and bottom surface. Both adlayers are terminated with carbon-oriented zigzag edges.

4. *In P9, the authors consider that the two triangular islands with different orientation angle grow on top and bottom surfaces respectively, because the growth is not interrupted when two islands overlap. But it cannot conclude that other observed Ti_5C_4 regions also result from overlapping of adlayers growing on both surfaces. The Ti_5C_4 regions in Fig. 2j can result from two layers of TiC on the same surface.*

R. We agree with the reviewer that the Ti_5C_4 region in Fig. 2j outlined by black dashed triangle might be from two layers of TiC on the same surface. As the black triangular area shows the same parallel edges as the outer adlayer, we cannot rule out the possibility that there are two layers at the surface. However, such growth is probably not common because otherwise we should expect to see large number of regions with Ti_6C_5 and Ti_7C_6 structures. In this special case, the lower-right edge of the adlayer happens to be one edge of the pore, which might be the reason why atoms could migrate directly onto the adlayer. We have thus tuned down our claim in the paper by replacing ‘strictly single-layer growth’ to ‘mostly single-layer growth’. We also discussed possibility of having two adlayers on one surface in the manuscript in Line 221, Page 10.

5. This work is so special, it looks like an extreme case of epitaxial growth of TiC on TiC substrate, the only thing is the thickness of the substrate is ultrathin then can be labeled as $Ti_{n+1}C_n$ ($n>1$); although the author can obtain small scale Ti_4C_3 and Ti_5C_4 because the substrate is Ti_3C_2 and the growth source is limited, it's hard to predict the case in large scale synthesis under continuous flux of source. In my opinion, the significant of this work seems not as big as the author claimed.

R. We agree with the reviewer that the work is special that the substrate is ultrathin. The work however for the first time proves the possibility of Frank van der Merwe growth of single layer transition metal carbides (TMC) and provides detailed explanation on the growth mechanism. This will certainly lead to increasing interest in optimizing growth parameters to achieve 2D TMC at larger scale. Although an ultrathin substrate seems to be difficult to achieve, many ultrathin 2D materials such as graphene, BN, and transition metal dichalcogenides exist in free-standing form up to several tens of microns, serving as substrates to grow other 2D materials (for example, monolayer MoS_2 on monolayer MoS_2 using CVD, Adv. Mater. 2018, 30, 1704674). Another possibility is to use, for example, hexagonal closed-pack (hcp) metal compounds or hcp metals (Sc, Ti, Zr, or Hf) with a hexagonal metal (h -M) surface layer with the lattice constant comparable to MXenes to serve as the substrates to grow hexagonal transition metal carbide layer M_nC_n . The surface M layer in the substrate could be more strongly bonded to the first C layer than to the rest of the substrate, which could be etched out, leaving the synthesized $M_{n+1}C_n$

MXene. So, there are possible ways that may scale up the growth mechanism observed here. We have added such discussion in the **Summary** of the manuscript. To not claim more than what we discovered, we have removed the statement about growing other 2D materials in the revised manuscript.

Reviewer #2

In this study, the authors used in-situ STEM technology combined with DFT and MD simulations to demonstrate a clear strategy for bottom-up homoepitaxial Frank-van der Merwe growth of 2D h-TiC on single layered Ti₃C₂ MXene. This is a piece of nice work with high novelty, in which new phenomenon is observed and the theoretical calculations explain the experimental observations very well. I would like to recommend its publishing in Nature Communications after the following comments are addressed.

R. We would like to thank the reviewer for the positive and useful comments.

1) The authors demonstrated that the separated triangle h-TiC layers in Fig. 2j belong to the top and bottom surfaces, respectively. However, the depth of focus in STEM mode is very small and the contrast of two h-TiC layers are normally different at the same defocusing value. STEM simulation would be helpful to confirm this speculation.

R. We agree with the reviewer that in principle the contrast of adlayers on different surfaces may have different contrast. The two adlayers on top and bottom layer have a depth difference of around 1 nm. STEM simulation was performed for two supercells with one adlayer on the top surface and another adlayer on the bottom surface with slight overlap (a-d) and without overlap (e-h) as shown in the figure below. The electron beam is focused on the top surface. From the line profile based on the simulated STEM image (Figure R3), we can see that there is slight difference in peak intensity for the two layers but such difference if observed experimentally is not convincing enough to support adlayer depth considering influence of noise on peak intensity in experimental STEM images. In the manuscript, we have now adopted the approach proposed

by Review #1 using the orientation of the adlayer triangle to tell if two adlayers are on different surfaces.

Figure R3. Top view (a) and perspective view (b) of the crystal structure model of the supercell used for STEM simulation with two partially overlapping adlayers on top and bottom surfaces, respectively. (c) The simulated STEM image using supercell shown in (a) and (b). (d) Intensity profile along the white arrow in (c). Top view (e) and perspective view (f) of the crystal structure model of the supercell used for STEM simulation with two non-overlapping adlayers on top and bottom surfaces, respectively. (g) The simulated STEM image using supercell shown in (e) and (f). (h) Intensity profile along the white arrow in (g).

Besides, it is possible that the two h-TiC layers locate on the same surface because when they meet together, the formed grain boundary (Fig. 2k) could act as the nucleation site, due to its low energy, to grow a new h-TiC layer on their top to from Ti5C4 regions.

R. The reviewer here proposed an alternative explanation to the captured dynamics in Figure 2j-m. Although it seems possible, we think such vertical growth at the grain boundary based on the hypothesis that the two islands are on the same surface is quite unlikely. First, the source atoms need to find a way to move to the grain boundary and then migrate up onto the adlayers.

Therefore, the growth of the adlayer will most likely start from the two corners of the grain boundary, which is contradictory to the experimental observation that the growth front is still parallel to the edges of both adlayers. Second, in case one grain grows above another grain, we should see expansion of only one grain while the other grain on the bottom should stop growing over the grain boundary. From Figure 2m we can see both grains have expanded passing the original boundary. Also, as pointed out by Review #1, the two grains have different orientations, which strongly support that they are on two different surfaces.

2) Why should the migration paths marked in Fig. 3b and 3f be the only way? Does a different migration path design influence the calculated energy barriers in the end?

R. The migration paths shown in Figure 3b-f are not the only paths. More paths with different configurations of surrounding defects have been explored in the Supplementary Figures S4-S7. Those paths show quite similar energy barriers compared to the paths shown in Figure 3. The paths shown in Figure 3 gives a general idea that defects can significantly reduce migration energy of Ti and C atoms onto the surface. We do not expect those paths are the only reasonable paths in reality. We have added a sentence to clarify this in the manuscript in Line 173, Page 8.

3) The authors should give the explanations on the terms “P, -S, -B, V1-F, V2-F, V1-N, V2-N” in Supplementary Figure S5-S7?

R. We would like to thank the reviewer to point this out. The terms are now explained in the figure captions of Supplementary Figure S5-S7.

4) The authors mentioned that $Ti_{n+3}C_{n+2}$ with a “ABCABC...” stacking configuration in analogy to rock-salt structured TiC is the most stable structure. Is there any relationship between the stability of $Ti_{n+3}C_{n+2}$ and the number of TiC layers?

R. Indeed $Ti_{n+3}C_{n+2}$ with a “ABCABC...” stacking resembles the (111) planes of cubic TiC rock-salt structure. The figure below shows that as number of Ti layer increases, the formation energy decreases. The cubic TiC has the lowest formation energy.

Figure R4. Formation energy of $Ti_{n+1}C_n$ MXene as n increases. The red line indicates the formation energy of bulk cubic TiC. All the formation energies are calculated with respect to hcp metal Ti and graphite carbon.

5) The growth method is a little complex because monolayer Ti_3C_2 substrate with h-Ti surface should be obtained first in order to grow single layer h-TiC. Furthermore, to study the properties and applications, it is necessary to transfer the as-grown samples onto target substrates. Could the authors give some discussions on these two points?

R. The current growth method is indeed quite complex, as also pointed out by Reviewer #1. However, there is possibility to use the concept here to grow 2D TMC on a different substrate. The Ti_3C_2 substrate has two main characteristics: the very small thickness and the *h*-Ti surface. Future study will try to find suitable substrates with at least one of the two characteristics. For example, many 2D materials such as graphene, BN, and transition metal dichalcogenides exist in free-standing form, serving as substrates to grow TMC. Secondly, numerous hexagonal or cubic metal compounds or metals with a hexagonal metal surface (*h*-M) with the lattice constant comparable to MXenes can serve as the substrates to grow M_nC_n . For example, the (0001) oriented Sc, Ti, Zr, and Hf metals have hexagonal metal surfaces and lattice constants comparable to MXenes. The surface M layer in the substrate could be more strongly bonded to the first C layer than to the rest of the substrate, which could be etched out, leaving the synthesized M_{n+1}C_n MXene. We are currently working on a follow-up theory paper on how to choose substrates with *h*-M surfaces based on bonding energies.

In our experiment, it is difficult to transfer the as grown sample as it is either over vacuum or already on the SiN substrate used in the heating chip. With other substrates that potentially could be used to grow TMC as discussed above, it might be possible to transfer the as grown TMC through etching out the substrate.

We have added discussion regarding the reviewer's comments in the manuscript in Line 335 Page 16.

6) About the universality of the growth concept, the authors pointed out that it would lead to bottom-up synthesis methods of various 2D materials beyond 2D TMC. Could the authors give more discussions on the materials that might be synthesized by this method?

R. After careful thought on this issue, we decided to remove this statement in the paper because currently we do not have sufficient theoretical or experimental evidence about what kind of other 2D materials could be grown using this method. We initially thought metal atoms with very low diffusion barrier could form 2D adlayer on suitable substrates, but it probably requires more evidence to come to such a claim in the paper.

Reviewer #3

This manuscript reports homoepitaxial growth of atomic layers on MXenes stimulated by electron beams. The detailed growth mechanism of TiC monolayer has been investigated by combining in-situ STEM, DFT calculations, and ReaxFF MD simulations, which synergically offer insights into the growth process. In addition, using the electron beam irradiation, the authors are able to obtain new MXenes that have not been synthesized before. I anticipate that this work will further stimulate investigations on the bottom-up synthesis of MXenes that are highly desired due to their potential applications. I therefore strongly recommend its publication, after the authors address the following issues:

R. We would like to thank the reviewer for the positive and useful comments.

1. Scale bars are missing at several places: Figure 1h and Figure 5.

R. We thank the reviewer for pointing this out. The scale bars have been added in Figure 1h and Figure 5.

2. What is the feasible range of chemical potential of Ti (Fig. 4, SI Figure 11-12)?

R. The range of chemical potential of Ti, $\mu(\text{Ti})$, depends on the form of Ti and C sources. If both sources are in bulk phases (the total energy of bulk Ti and C is -7.76 and -9.22 eV/atom, respectively), then $\mu(\text{Ti})$ ranges from -9.30 to -7.76 eV, which corresponds to the Ti poor (C rich) and Ti rich (C poor) conditions. This is the range we showed in the main text. If Ti and C are in atomic form (the total energy of Ti and C atom is -2.45 and -1.37 eV, respectively), then the range of $\mu(\text{Ti})$ is much broader from -17.15 to -2.45 eV. This equals to $\Delta\mu$ from -6.81 to 7.89 eV as partly shown in the supplementary Figure S12 and S13.

3. Would the edges of the islands be different from the edges of pores?

R. The edges of the pores are more complicated than the edges of the islands because the edges of the pores have three Ti atom layers while the adlayers are strictly single layer. We can tell that

the edges of pores are also aligned along $\{10\bar{1}0\}$ planes, similar to the adlayers. From this aspect, maybe those edges can be denoted as zigzag edges, although the concept of zigzag edges has not been used to describe edges with five atom layers in 2D materials system. However, as the paper mainly focuses on the growth mechanism of the 2D adlayer, the edge structure of the pores is not discussed in detail in the current manuscript. We have added a brief description of the pore edges in the section '**In situ Homoeptaxial Growth**', see Line 109, Page 5.

4. In the DFT calculations, the authors used the approaches established by Liu and Yakobson et al to study direction-dependent edge energy and growth kinetics. Please credit those papers: PNAS, 2012, 109 (38) 15136-15140, Phys. Rev. Lett. 2010, 105, 235502. Also, recently MXenes have been demonstrated to be promising electrodes for 2D electronics (J. Am. Chem. Soc., 2016, 138 (49), pp 15853–15856), it would be informative to the readers if the authors can discuss this in the introduction.

R. We thank the reviewer to suggest relevant papers and have referenced them accordingly. We have added discussion of application of MXenes as electrodes for 2D electronics in the introduction in Line 43, Page 2.

Reviewers' Comments:

Reviewer #1:

Remarks to the Author:

The authors have answered all my comments in a satisfactory way and I am pleased to recommend this manuscript for publication.

Reviewer #2:

Remarks to the Author:

The authors have addressed all my comments properly, I would like to recommend the publication in the present form.

Reviewer #3:

Remarks to the Author:

The authors have addressed my previous comments and i recommend its publication.

We would like to thank the reviewers for agreeing to publish the paper.

Reviewer #1 (Remarks to the Author):

The authors have answered all my comments in a satisfactory way and I am pleased to recommend this manuscript for publication.

Reviewer #2 (Remarks to the Author):

The authors have addressed all my comments properly, I would like to recommend the publication in the present form.

Reviewer #3 (Remarks to the Author):

The authors have addressed my previous comments and i recommend its publication